The reuse of public datasets in the life sciences: potential risks and rewards

http://orcid.org/0000-0002-4022-8531 Sielemann Katharina 1 2 kfrey@cebitec.uni-bielefeld.de
http://orcid.org/0000-0003-4262-9176 Hafner Alenka 1 3
http://orcid.org/0000-0002-3321-7471 Pucker Boas 1 4
1 Genetics and Genomics of Plants, Center for Biotechnology (CeBiTec) & Faculty of Biology, Bielefeld University , Bielefeld , Germany
2 Graduate School DILS, Bielefeld Institute for Bioinformatics Infrastructure (BIBI), Bielefeld University , Bielefeld , Germany
3 Current Affiliation: Intercollege Graduate Degree Program in Plant Biology, Penn State University , University Park, State College, PA , United States of America
4 Evolution and Diversity, Department of Plant Sciences, University of Cambridge , Cambridge , United Kingdom
Lortie Christopher
Electronic publication date: 2020 Sep 22
Publication date: 2020
Volume: 8
Electronic Location ID: e9954
Received 2020 Apr 30; Accepted 2020 Aug 25
Copyright: © 2020 Sielemann et al.
Copyright year: 2020
Copyright holder: Sielemann et al.
License: This is an open access article distributed under the terms of the Creative Commons Attribution License, which permits unrestricted use, distribution, reproduction and adaptation in any medium and for any purpose provided that it is properly attributed. For attribution, the original author(s), title, publication source (PeerJ) and either DOI or URL of the article must be cited.
License URL: https://creativecommons.org/licenses/by/4.0/

Keywords: Reuse, Data science, Sequencing data, Genomics, Bioinformatics, Databases, Computational biology, Open science

Funding: Deutsche Forschungsgemeinschaft Bielefeld University St. Catharine’s College, University of Cambridge Support for the Article Processing Charge is provided by the Deutsche Forschungsgemeinschaft and the Open Access Publication Fund of Bielefeld University. K.S. is funded by Bielefeld University. A.H. received the 2018 Richard Hardy Award (St. Catharine’s College, University of Cambridge) which partly supported an internship at Bielefeld University, leading to this collaboration. The funders had no role in study design, data collection and analysis, decision to publish, or preparation of the manuscript.

==============================
The ‘big data’ revolution has enabled novel types of analyses in the life sciences, facilitated by public sharing and reuse of datasets. Here, we review the prodigious potential of reusing publicly available datasets and the associated challenges, limitations and risks. Possible solutions to issues and research integrity considerations are also discussed. Due to the prominence, abundance and wide distribution of sequencing data, we focus on the reuse of publicly available sequence datasets. We define ‘successful reuse’ as the use of previously published data to enable novel scientific findings. By using selected examples of successful reuse from different disciplines, we illustrate the enormous potential of the practice, while acknowledging the respective limitations and risks. A checklist to determine the reuse value and potential of a particular dataset is also provided. The open discussion of data reuse and the establishment of this practice as a norm has the potential to benefit all stakeholders in the life sciences.

Introduction

Data reuse as a part of the ‘big data’ revolution

Data reuse is an essential component of open science and has been facilitated by the ‘big data’ revolution. The transition from (hand) written notes to datasets stored on hard drives can be viewed as the first step on the road to effective data reuse in the life sciences (Fig. 1), allowing the generation of multiple copies at almost no additional cost. The second step was improved connectivity, which was enabled by the internet. Together, these technological advances in data storage and transfer enabled a worldwide exchange of ‘big data’, which is common in biology (e.g. sequence data). This technological basis made data sharing possible (Fig. 1). Next, it needs to become convenient for researchers to share data through increased accessibility. Obligations and benefits lead to established sharing behavior, which results in more datasets becoming available, which can then be reused in turn. Finally, it becomes common practice and a habit to share all data, resulting in a positive feedback loop. Data sharing is already common in several disciplines, including genomics, neuroscience, geoscience and astronomy, and an increasing number of studies reuse shared data (Pierce et al., 2019; Tenopir et al., 2020).

Figure 1 The evolution of data sharing behaviour.

(1) Technical progress makes global sharing of large datasets possible, (2) increased accessibility to required technology makes it widely available, (3) obligations and benefits for researchers establish sharing behaviour, (4) the size of datasets increases and makes them attractive, (5) reuse develops over time—which results in a positive feedback loop and a habit to share all data.

For clarity purposes, we distinguish here between fair reuse (for novel purposes, e.g. meta-analysis), reproduction of previous studies with available data (a vital component of open science), and unjust reuse (dual publication and plagiarism). Alongside the reproduction of studies to test findings, only fair reuse should be considered a valid component of open science and is the type of reuse our discussion focuses on.

To establish data sharing as a norm it had to be introduced through obligations and promoted through benefits for researchers (McKiernan et al., 2016). Numerous funding agencies, publishers (e.g. Nature: (Announcement, 2016), NSF: (‘Biological Sciences Guidance on Data Management Plans’, https://www.nsf.gov/bio/biodmp.jsp), PLOS: (‘PLOS ONE—Data Availability’, https://journals.plos.org/plosone/s/data-availability)) and private foundations require all data be made publicly available within a certain time frame, with an indication that this leads to increased transparency in the field (Parker, Nakagawa & Gurevtich, 2016; Tenopir et al., 2020). Many international data sharing guidelines like Findable Accessible Interoperable Reusable (FAIR) (Wilkinson et al., 2016), TOP (The Transparency and Openness Promotion) (Nosek et al., 2015), Open Data in a Big Data World (Open Data in a Big Data World, 2016) and the Beijing Declaration (CODATA, 2019) have emerged by the necessity of the ‘big data’ revolution. The sharing of datasets leads to statistical robustness and allows re-analysis of existing datasets underlying authors’ claims (Open Data in a Big Data World, 2016), while enabling the discovery of novel patterns through meta-analysis (Duvallet et al., 2017). Such data reuse leads to cost reduction, reproducibility and accountability of research, scientific discovery and detection of novel biological information, and can also be helpful in other areas like education, business and government (Safran, 2017; Pasquetto, Randles & Borgman, 2017; Porto, Pires & Franco, 2017; Leonelli et al., 2017).

Despite these measures and benefits, data reuse is still not ubiquitous, for which there may be different explanations. In 2017, an analysis of 318 biomedical journals revealed that only 11.9% of them explicitly stated that data sharing was required as a condition of publication (Vasilevsky et al., 2017). Additionally, a survey of 100 datasets associated with ecological and evolutionary research showed that 56% of the databases were incomplete, and 64% were archived in a way that partially or entirely prevented reuse (Roche et al., 2015). Thus, if the publicly available datasets are not widely re-examined (either checked for quality and/or re-analysed), enforcement of open science through policy may not be sufficient to harness the full power of global sharing (Pasquetto, Randles & Borgman, 2017). The main causes of researchers refraining from reusing publicly available datasets are (1) concern about the quality and reliability of data (often warranted), (2) a lack of awareness about the potential in big data or (3) insufficient bioinformatics knowledge to mine the data (Denk, 2017). Regardless of the cause, the resulting ‘backlog’ of under-utilised reliable datasets leads to unnecessary experiments (e.g. extensive repetitive sequencing increasing costs) and likely hides useful undiscovered patterns.

Types of reusable data

There are numerous different types of datasets which harbour reuse potential (Fig. 2). These include but are not limited to (1) publications which are accessible to text mining; (2) sequences of genomes, single genes or plasmids or whole sets of sequence reads, (3) annotations of sequences (e.g. sequence motifs collected in JASPAR (Sandelin, 2004)), (4) chromatography results and mass spectra, (5) information about the structure of proteins, (6) biochemical parameters of enzymes (e.g. affinity or reaction speed), (7) geodata (e.g. coordinates of observations), (8) images of biological material or geographical regions as well as plots and diagrams, (9) algorithms and software (e.g. code), measurements of automatic sensors and global-scaled observatory networks (Soranno et al., 2015), and (11) phenotypic data (Arend et al., 2016).

Figure 2 Types of reusable data classified into primary and derived/secondary data.

Specific examples for each data type are provided in parentheses. The data classification is based on: Wooley & Lin, 2005. (Sources of the pictures: ‘Protein Data Bank in Europe—Logo’, https://www.ebi.ac.uk/pdbe/about/logo; Pucker, Holtgräwe & Weisshaar, 2017; Schilbert et al., 2018; Mitchell et al., 2019; Frey & Pucker, 2020).

Generally, datasets can be classified as primary and derived. Primary can be defined as including direct, experimentally obtained data’ and derived as including meta-analyses and processed results. Statistical data is an example for the fact that data cannot always be classified into ‘primary’ and ‘derived’. Results of a specific experiment can be directly statistically evaluated (primary), whereas results of multiple studies can be assessed and compared in a statistical manner too (derived). Further, it has to be considered that specific fields require the integration of data of various types, formats and abundance (Leonelli et al., 2017), which is hard to achieve by a single database, and therefore requires cooperation to encourage data reuse.

Purpose of the review

Education about the opportunities, challenges, limitations and techniques of data reuse is an important topic that has not been sufficiently addressed in the literature. This review aims to provide an entry point to the discussion of data reuse as part of open science, accessible to a wider audience of life scientists, not only bioinformaticians. The benefits of data sharing (and reasons for refraining from it) have been extensively reviewed in fields other than bioinformatics, for example ecology (Hampton et al., 2013; LaDeau et al., 2017), medicine (Wade, 2014; Krumholz, 2014; Safran, 2017; Hulsen et al., 2019) and cell biology (Dolinski & Troyanskaya, 2015). However, the existing reviews do not summarise the common benefits and challenges that arise from data reuse in the life sciences.

Here, we highlight the potential of data reuse as well as hurdles which need to be overcome. The presented benefits, challenges, risks and potential solutions range across different fields and aim to illustrate the inherent characteristics of data reuse. We define ‘successful reuse’ as the use of previously published data to enable novel scientific findings, usually resulting in a peer-reviewed publication. As showing case studies could reduce the initial barrier to reuse by demonstrating its value in a more practical manner (Curty et al., 2017), we use examples of successful reuse of different data types (genome, transcriptome, proteome, metabolome, phenotype and ecosystem) to illustrate its enormous potential. In addition, we provide a checklist of questions for biologists without extensive experience in handling public datasets to consider when determining whether a particular dataset is fit for reuse. We focus on the reuse of primary data, especially different sequences (as the data type characteristic of the life sciences and continuously producing vast amounts of information). The benefits for individuals and the scientific community as a whole make a strong case for reuse of data but only when the risks and limitation have been taken into account.

Survey Methodology

We performed a literature review to draw an informed picture of the benefits and challenges associated with data reuse in the life sciences. The publication survey was performed on 14th June 2020, using the PubMed databases to search for relevant peer-reviewed journal articles. The publication year and type of journal were unrestricted. We entered the following search term: ‘data reuse’ (Title/Abstract) OR ‘dataset reuse’ (Title/Abstract). PubMed produced 188 results (Table S1). Articles from health science, clinical research and medicine were not considered, unless they were reviews. Databases were also excluded, leaving 25 papers (Table S1). From the sources we located in our non-systematic survey, recurring benefits, potential, limitations, challenges and risks of data reuse were identified and categorised, in order to make them more reader friendly. Where we found the issue not sufficiently explained in the article resulting from the initial search, additional resources were sought on that particular topic (on PubMed using the relevant keywords or directly from articles cited in that particular paper). All resulting references are cited in ‘Potential of reusing public datasets’, ‘Challenges, limitations and risks of data reuse and possible solutions’, ‘Research integrity considerations’, and ‘Recognizing the value of data reuse’ sections.

Next, we explored published examples from different fields in the life sciences where data reuse was performed. In order to find examples from a range of biological disciplines, we first identified different types of data that can be reused in the life sciences (genome, transcriptome, proteome, metabolome, phenotype, ecosystem). The relevant literature was selected through the authors’ experience in bioinformatics, genetics and genomics of plants, and plant molecular biology. We combined some more specific terms with the keywords mentioned above, namely ‘genomics’, ‘bioinformatics’, ‘sequencing reads’ and ‘transcriptomics’ as well as search term keywords specific for each row in the table including the examples for performed data reuse, for example, ‘coexpression’, ‘pangenomics’, ‘network analysis’ and ‘metagenomics’. We acknowledge this is primarily a selection of successful instances of data reuse that focuses on sequences, with only selected examples from other life sciences, however, we deem it illustrative of the potential of data reuse, which was our aim.

To construct the checklist for the selection of datasets appropriate for reuse, we chose criteria based on the challenges and limitations identified through the literature review. The questions to consider, possible controls and suggestions were identified through our personal experience and backgrounds in bioinformatics.

Potential of Reusing Public Datasets

Reuse of publicly available data is strongly connected to numerous advantages, not only affecting the scientific community and society but also authors themselves and is generally viewed positively among researchers (Tenopir et al., 2020). Data sharing is the prerequisite for easy availability of reusable data, leading to positive reuse behaviour and can be key to improved integrity, transparency and reproducibility (Curty et al., 2017; Tenopir et al., 2020). Below is a collation of common benefits of reuse, though we acknowledge there are other advantages specific to particular fields and/or data types not listed here. By and large, information loss can be prevented, scientific knowledge can be expanded, authors can profit through higher reputation, and even databases can benefit from data reuse.

Preventing information loss

Making data publicly available for reuse is an elegant way to prevent information loss stemming from different issues. Data can be subject to irretrievable loss in case of storage solely on private computers and servers, which may not convert the data to the currently used format and that are subject to failure. This is avoided when data are shared with the public and stored in adequately funded databases with backup mechanisms. There are already numerous public repositories for genomic and gene-expression data, such as the Sequence Read Archive (SRA)/European Nucleotide Archive (ENA) and Gene Expression Omnibus (GEO), respectively. Recently, GEO has been used in a case study to improve dataset reusability with a literature recommendation system (Patra et al., 2020) and is highly recommended over other databases for the submission of RNA-Seq datasets (Bhandary et al., 2018). In medical research, information loss stems from large amounts of gathered data remaining inaccessible for reuse by a wider audience (sometimes even the authors) after the initial publication (Wade, 2014). Moreover, the development of new tools and methods leads to the possibility of extracting more information from a given dataset than was feasible at the time of publication. A notable example of such extraction of new information, is the basecalling step when working with nanopore sequencing data derived from Oxford Nanopore Technologies devices. Enhanced algorithms allow higher accuracy or even the identification of DNA modifications (Liu et al., 2019a, 2019b). Further, meta-analyses (e.g. the prediction of specific genomic features) and machine learning approaches require large amounts of data which are already available, and therefore should not and cannot be produced again. This is especially important in fields where data is complex, like videos in organismal biology (Brainerd et al., 2017). Such examples illustrate the importance of sharing and maintaining existing datasets, alongside supplementing them with new data, in order to prevent information loss.

Expanding scientific knowledge

Reuse of data from different sources holds immense potential for scientific discovery and therefore generally enhances scientific progress (Curty et al., 2017). Integration of data from different sources, for example, of exRNA metadata, biomedical ontologies and Linked Data technologies can facilitate interpretation and hypotheses generation by providing independent biological context (Subramanian et al., 2015). In medicine, reuse of data collected as a by-product of health care has the potential to transform the practice of medicine and its delivery, which is a compelling argument of reuse benefits outweighing the risks (Wade, 2014; Safran, 2017). For example, reuse can help eliminate bottlenecks in biomedical research at all translational levels and data-mining (the hypothesis-free search for patterns in data) can reveal potential starting points for experimental medical research (Wade, 2014). Another method of reuse is meta-analysis, which can elucidate new patterns and produce novel hypotheses—inaccessible from the analysis of any individual dataset. In turn, gene expression studies generating large datasets tend to look for general patterns and thus an in depth inspection of single genes can provide additional insights (Bhandary et al., 2018). Temporal and spatial limitations of a single experiment can be overcome by new combinations of existing data and applications integrating different disciplines can be made possible (Curty et al., 2017), leading to more interdisciplinary collaboration (Tenopir et al., 2020). When examining communities in ecology, metagenomics, metatranscriptomics, metabarcoding, and metaproteomics provide insights into community composition and function (Ten Hoopen et al., 2017). Crucially, publicly available data can lead to the development of new experimental designs (Grace et al., 2018) and can be connected with complementary knowledge and reused in novel experiments (Martens & Vizcaíno, 2017) that contribute to expanding scientific knowledge.

Maximising time, labour and cost efficiency

Reuse of data saves time and money, thus is more economical, and can offer the opportunity to overcome the restraints of limited experiments, high costs and technical difficulties (Raju, Tsinoremas & Capobianco, 2016; Curty et al., 2017). In medicine, the reliance on expensive experimental research when a wealth of existing data is available has been criticised and reuse of existing data and information was presented as a solution (Wade, 2014). In metagenomics, where datasets tend to be in the gigabyte range, appropriate archiving of workflow intermediates for reuse can decrease the costs of re-analysis (Ten Hoopen et al., 2017). Additionally, many datasets deposited in sequence repositories like GEO were collected at enormous effort and used only once and so reusing them greatly increases their utility (Patra et al., 2020). The labour efficiency of reuse is also illustrated by the famous eGFP browser (Winter et al., 2007) which presents the content of RNA-Seq datasets to biologists in a simple way. The alternative would be downloading and analysing raw RNA-Seq datasets from the SRA, requiring a substantial amount of bioinformatics expertise and computational power during the analysis. Valuable computational resources are provided by international and national cloud computing services like Elixir, CyVerse or the German Network for Bioinformatics Infrastructure (de.NBI) as well as by commercial organisations. Data reuse is a profitable choice for researchers to lessen expenses and shorten the research process as data collection was already performed by others (Curty et al., 2017). Cutting such costs through reuse (Fell, 2019) enables groups with small budgets to harness extensive datasets and thus enhancing equality.

Benefits for authors

While the scientific community and society are profiting most from public datasets, there are additional benefits of open access for authors themselves (McKiernan et al., 2016; Leitner et al., 2016; Ali-Khan, Harris & Gold, 2017). Researchers can build a reputation by generating high quality and well-documented datasets. Although compliance with data standards might be seen as an additional burden in some cases, the chances of reuse occurring are increased by providing data in the proper format (Rocca-Serra et al., 2016). Dataset sharing may also increase visibility of the associated research and results in additional citations; an added encouragement for authors (Piwowar, Day & Fridsma, 2007). There are even reuse examples which are possible without biological context, like benchmarking of bioinformatic tools or the identification of patterns (Bhandary et al., 2018). It has been shown that there is a robust, statistically supported citation benefit from ‘open data’ in comparison to similar studies without publicly available data (Piwowar & Vision, 2013). This especially aids early career researchers, who are outsiders of the scientific establishment and likely experience more barriers to other aspects of open science (National Academies of Sciences, Engineering, and Medicine, 2018), yet are highly involved in data collection and analysis (Farnham et al., 2017).

Benefits for databases

The reuse of sequence data is of increasing importance due to the large and rapidly growing size of the databases storing them (Fig. 3). The size of the SRA alone increased from 3,092,408 entries to 6,243,265 entries within 2 years (September 2016–September 2018) (NCBI Resource Coordinators, 2017; Sayers et al., 2019a). This growth rate continues to increase exponentially (Fig. 3). GenBank comprises a total of 3,677,023,810,243 sequences (2018) with an increase of 39.52% in comparison to 2017 (Sayers et al., 2019b). Approximately 120 million sequences and annotations of proteins were available within UniProtKB/TrEMBL in 2018 (The UniProt Consortium, 2019). Since managing an exponentially growing database has numerous challenges (Lathe et al., 2008), it is important to consider how they can be addressed through changing practices, including data reuse.

Figure 3 The increasing size of selected databases over time.

The number of bases/sequence entries in GenBank, the Sequence Read Archive (SRA) and UniProtKB/TrEMBL are shown, respectively. Note the logarithmic scale of the y-axes. The drop of sequence entries in UniProtKB/TrEMBL (in 2015) can be explained by the removal of duplicates.

The issue of the rate of nucleotide and proteomics data generation growing faster than storage capacity (Cook et al., 2016) can also be partially addressed through data reuse. Reusing available data instead of producing new and redundant datasets (i.e. when a large number of datasets that are in consensus is already available) results in a lower number of duplicates (Grace et al., 2018) and keeps databases concise. Since it takes an enterprise to maintain and upgrade the largest and most-used databases (meaning they must be adequately financed to survive), limiting redundancy through reuse is beneficial. One example of a sustainable business model is The Arabidopsis Information Resource (TAIR) database (Lamesch et al., 2012) which is funded by subscriptions from academic and non-profit institutions but allows a limited number of accesses by individuals. The public availability of datasets also allows the development of effective algorithms to tackle the bottleneck of data processing, all without the need to perform any sequencing (i.e., uncoupling the problem from access to sequence technology and allowing participation of e.g. computational fields). Additionally, not only the storage of a large number of datasets but also the actual reuse of the available data might increase funding of public databases and therefore ensure the long-term existence of these infrastructures.

Challenges, Limitations and Risks of Data Reuse and Possible Solutions

As discussed above, open access to datasets and studies would accelerate science while being cost-efficient (Spertus, 2012). However, it is important that the limitations of particular datasets are identified and the associated risks assessed. This is of particular concern in clinical trials as the results could have a direct impact on the patients involved, for example in the case of invalid secondary analyses which might harm public health (Sharing Clinical Trial Data, 2015). Here, we discuss selected common disadvantages and acknowledge that the reuse of specific data types possesses some field-specific issues not addressed here (e.g. adding to the burden of paperwork in clinical medicine (Safran, 2017)).

Unknown quality

For successful reuse appropriate data quality is required in order to be reliable for the user. There are inherent quality differences between (1) user-submitted public datasets, (2) carefully curated databases for specific organisms, (3) ones with inherently small holding sizes (like PDB or SwissProt) and (4) phenotype databases. Additionally, information regarding experimental design, methods and conditions is often incomplete and results in datasets unsuitable for reuse. Mislabelled or swapped samples alongside intrinsic errors, like missing technical replicates, pose a problem that is almost impossible to identify when accessing a public dataset. Precise documentation of the workflow is crucial to clearly indicate limitations of the generated data product (Soranno et al., 2015). Despite this, the peer-review process rarely reaches the datasets and their descriptions (Patra et al., 2020). There is also a trade-off between the collection of detailed metadata during submission and high submission numbers. However, additional requirements for data submission should not result in fewer publicly available datasets (Rung & Brazma, 2013). Ultimately, the limitations of each study (and dataset) are best known by the primary investigators and not by the community accessing the data—a trade-off that studies based on reused data must consider.

It is also difficult and time-consuming for the user to discover available data which is relevant and suitable for the analysis and further to ensure sufficient quality of these datasets (Curty et al., 2017). Even if complete metadata are provided, accessing numerous different webpages to collect all information associated with a combined dataset can be tedious (Bhandary et al., 2018). This leads to a risk of wasted time and effort on flawed data, thus requiring trust in data producers and their methods and techniques (Curty et al., 2017). Moreover, simply using a large amount of publicly available datasets does not inherently lead to correct patterns. Despite the importance of trends revealed from large datasets, it is not necessarily the case that a large number of reads/replicates, with an associated low noise, means that the emerging trend is true. Conversely, one can imagine a trend with low noise and low deviation, produced from a large dataset, but with the data coming from one author/group that has an undetected systematic error. Equally, low noise and the use of a small dataset that is believed to be of ‘higher quality’ (i.e. has been thoroughly checked for errors), may hide a trend or show a nonvalid one. Ultimately, the low availability of necessary metadata in standardised formats, with insufficient additional information leads to a lack of reproducibility and can result in misuse and wrong assumptions (Rung & Brazma, 2013; Curty et al., 2017).

Denormalization

Of particular concern is the circular reuse of data. It can, for instance, lead to heavily denormalized annotation in databases, i.e., the same data is stored multiple times in the same database under different names/identifiers (Bell & Lord, 2017). When such data duplication is not recorded and the user is not made aware of it, the data distribution in the database does not reflect the true data distribution. For example, sequencing and annotation errors can be propagated by reuse and not eliminated by additional published sequences that would reveal it to be statistically insignificant. For annotations, it has been shown that it is possible to detect low-quality entries (resulting from this denormalization) by looking for specific patterns of provenance in the database (Bell, Collison & Lord, 2013). With respect to gene models, this problem could be addressed in the future by integrating RNA-seq datasets in the annotation of new genome sequences. In terms of functional annotations, this issue persists because the experimental characterisation of numerous genes in a diverse set of species cannot be expected in the near future.

Comparison and integration of datasets and databases

The comparison and integration of datasets from different sources remains a challenge of reuse (Pasquetto, Randles & Borgman, 2017). In metagenomics, when communities (that have been studied independently) are compared several issues arise, including differences in workflow, unrecorded variables, non-unified presentation format, and relevant raw data not being publicly available (Ten Hoopen et al., 2017). The same issue is illustrated by the enormous differences in annotation provided by the different databases, for example on NCBI (Genome), ENSEMBL, and Phytozome for the same species. In plant phenotyping, reuse and meta-analysis is challenging as data comes from different experimental sites, plant species and experimental conditions, while including many different data types (Papoutsoglou et al., 2020). This non-comparability is also repeated in medical research (Wade, 2014). Therefore, for any valid comparison between datasets from different databases or for integration of databases themselves, conditions specific to the data type and field have to be satisfied.

Different file types and data structures pose a universal challenge for integration which can be overcome by the use of standards if these are de facto accepted by the community (Rocca-Serra et al., 2016). For communication between disciplines, an interdisciplinary team with ‘brokers’ has been recommended for the setup of new databases (Soranno et al., 2015). Consistent use of controlled vocabularies and standardised languages like XML (Soranno et al., 2015) also enhances the value of data collections substantially. For example, in meta-analysis of sequences an established and unified file standard is crucial (Ten Hoopen et al., 2017). FASTA (Pearson & Lipman, 1988), FASTQ (Cock et al., 2010), and SAM/BAM (Li et al., 2009) are famous examples of file standards that have allowed the effective exchange of information between numerous groups involved in the earliest sequencing projects (Leonard & Littlejohn, 2004; Ondřej & Dvořák, 2012; Zhang, 2016). Any disparities in the sampling method also have to be taken into account when biological material is concerned, so it is essential they are recorded appropriately (Ten Hoopen et al., 2017). When appropriate, unified workflow reporting standards (like described by Ten Hoopen et al., 2017) within a field would largely remove these inhibitions to reuse. It is for these reasons that guidelines like FAIR, which (Wilkinson et al., 2016) promote universal metadata standards across-the-board, are essential to allow comparison and integration of datasets.

Re-analysis as a possible solution

Re-analysis of publicly available data is one way to tackle the issues of unknown data quality, data denormalization, and database integration which arise with reuse. This can be achieved through curation and self-correction, with both being difficult to directly enforce. In the same manner that the re-examination of public biodiversity data leads to error correction (Miller et al., 2015; Zizka et al., 2019), so should sequence repositories reflect changes in the field’s consensus (e.g. on specific gene annotations). A way to address some of the risks of reuse through re-analysis would be investing in a controlled environment containing extensively peer-reviewed datasets (Spertus, 2012) and manually-curated databases. A defined, suitable environment or database could also include follow-up data for a detailed understanding of the primary data and the corresponding results. Further, re-analysis and reproducibility might be improved by conventions for standardisation, documentation and organisation of analysis workflows (Lowndes et al., 2017). This includes detailed records using open science tools like shared GitHub repositories (Lowndes et al., 2017). Crucially, reuse cannot occur if produced datasets are not widely released to public. Re-examination of databases, like that by Grechkin, Poon & Howe (2017) of SRA and GEO, to automatically identify datasets overdue for release are vital in this effort.

Re-analysis has proven to be efficient with some data types but is not practical in all cases. An excellent example of how investing in manually curated databases eliminates many issues are ‘expression atlases’ with annotated sequences checked for quality and re-analysed using standardised methods (Kapushesky et al., 2010). However, regarding the enormous and still increasing amount of sequence data, this is hardly an option for all data types. An analysis of reused data types already indicates that studies often rely on previously identified differentially expressed genes or calculated gene expression values instead of processing raw data again (Wan & Pavlidis, 2007). Therefore, different strategies might work for different data types or different communities. In all cases, specific standards and formats for data reuse should be applied (Pasquetto, Randles & Borgman, 2017), lest “the wealth of data becomes an unmanageable deluge” (Parekh, Armañanzas & Ascoli, 2015).

Metainformation as a possible solution

The trade-off between public access and unknown quality can be partially resolved by the publication of metadata (the information about the acquisition, processing and presentation) associated with a particular dataset. So far, most researchers (almost 60%) share their data and metadata using institution specific standards only or even without general metadata standards at all (Tenopir et al., 2020). Further, the metainformation necessary to make data findable (Tenopir et al., 2020) and to enable successful reuse differs between data types and between fields (Parekh, Armañanzas & Ascoli, 2015; Brainerd et al., 2017; Ten Hoopen et al., 2017; Papoutsoglou et al., 2020). The amount of metadata that can feasibly be recorded also varies by field, for example, the metadata about a single blood test on a patient includes countless variables (Safran, 2017).

Submitters need to be aware of the importance of providing accurate and complete metadata, but also that a controlled vocabulary is required to facilitate automatic identification of relevant studies/samples (Bhandary et al., 2018). People accessing the dataset also need to be aware of missing information about a dataset—an issue that can be resolved by including a ‘completeness of metadata’ search criterium in database search engines, like implemented at NeuroMorpho.org (Parekh, Armañanzas & Ascoli, 2015). The MassIVE Knowledge Base aggregates proteomics data including statistical controls and records of the data origin to ensure high quality of the datasets (Doerr, 2019) and is thus an example for a database providing data fit for reuse. The recently updated MIAPPE metadata standard for plant phenomic databases is another example of reuse facilitation through metadata formatting and was developed to address the shortcomings of the previous guidelines preventing FAIR-complying reuse (Papoutsoglou et al., 2020).

Additionally, methods for reconstructing existing databases are already being investigated. This can be done by curating existing metadata or extracting more of it through natural language processing techniques (Patra et al., 2020) and through metadata predicting frameworks (Posch et al., 2016). Many sequence databases, such as ENA, are handling this elegantly and submitting users can provide a very basic set of metainformation or provide comprehensive details about their study. There are also easy-to-follow instructions for the submission process (European Nucleotide Archive (ENA), 2020). Despite such incentives, many datasets still lack descriptions that would allow them to be re-sorted according to their metadata (Patra et al., 2020).

Publication of metadata is a practice already routinely implemented in data papers and data journals. Data papers (already common practice in Astronomy (Abolfathi et al., 2018)) have been indicated as a solution to the quality-check problem (Chavan & Penev, 2011) of reuse by providing descriptions of methods for collecting, processing, and verifying data (Pasquetto, Randles & Borgman, 2017). Widespread publication of such metadata in data journals (Figueiredo, 2017) is vital to the construction of high quality, peer-reviewed datasets. Consequently, data-focused journals, for example GigaScience, Scientific Data, and F1000, emerged during the last years. As long-read sequencing became affordable and paved the way for numerous high continuity assemblies, genome announcements describing new genomic or transcriptomic resources became popular. These publication types provide an elegant solution for data reports if a valuable dataset should be shared with the community but does not meet all criteria for publication as a full research article.

Research Integrity Considerations

Ethical considerations

Ethical considerations of data reuse are important to inspect in all life sciences fields and have been previously discussed in the context of clinical studies (Wade, 2014; Safran, 2017). Informed consent may not have been given with the knowledge that personal data will be utilised in more than the primary study. When sharing medical data, patients must not be identifiable, even if advanced methods are applied. This can require modification or masking of the dataset elements, for example defacing of brain images (Milchenko & Marcus, 2013). Similarly, genomic data (a part of modern medicine) is not covered by HIPPA and a parent’s genomic data could create a privacy risk for their children (Safran, 2017). Thus, there are ethical considerations specific to reuse of data in human research.

Research ‘parasitism’

The use of the same dataset in several different studies by the same author could be considered a type of dual publication in some circumstances (Beaufils & Karlsson, 2013). However, such reuse is not contentious to the same extent as plagiarism if it reveals novel findings and is not reused only to boost the number of publications. The trend of publishing from publicly available data (‘The parasite awards—Celebrating rigorous secondary data analysis’, https://researchparasite.com/; Longo & Drazen, 2016; Pucker & Brockington, 2018; Frey & Pucker, 2020) points to the crux of the matter of research integrity reservations about data reuse that some have. At the far end of this spectrum, there are authors exclusively using publicly available data (not generating their own to cross-check the quality), often choosing research topics/systems-of-interest based on the quality of data and not vice versa. This practice is associated with numerous advantages, including intensified use of existing datasets which effectively increases the ratio of value drawn from it compared to the costs of generating it in the first place. While multiple studies can benefit from reuse, long term risks might include funding bodies expecting reuse and rendering the acquisition of financial support for new experiments more challenging. Despite some expressed concerns regarding such ‘research parasitism’ (Longo & Drazen, 2016), including the fear of exploitation when acquiring the data was particularly expensive or labour-intensive (National Academies of Sciences, Engineering, and Medicine, 2018), the practice of reuse seems to prevail in the open science culture. The above-mentioned ‘Parasite awards’ use this tongue-in-cheek name to reward such practices, after the term was introduced to reprove of them. Due to the numerous benefits of reuse for the scientific community, we believe the term ‘research parasitism’ is unwarranted when fair reuse has been employed.

Recognising the value of data reuse by recognising the data producer

As perceived efficacy and efficiency of data reuse strongly influence reuse behaviour, it could be encouraged by demonstrating its value (Curty et al., 2017). So far, there have been extensive efforts to promote and develop standards for data sharing, but less effort to show the real value of data sharing or to recognise, cite or acknowledge the contributions of data sharing (Pierce et al., 2019; Tenopir et al., 2020). Adequate recognition of data producers would accelerate data sharing by eliminating the main barrier: the need to publish first (Tenopir et al., 2020).

Both data providers and creators of databases deserve recognition for their work. To a certain degree, this issue can be tackled with data publications, which might also prevent the splitting of a coherent dataset over multiple publications. Soranno et al. (2015) recommend specifically to describe the content of new databases in one paper with all contributors listed as co-authors and a description of the methods for the database development in an additional publication with all researchers involved in the process as co-authors. This is a practice that shares all its benefits with data papers (discussed above). The preservation and documentation of data provenance is crucial in aknowledging the support of data providers (Soranno et al., 2015). By supplying a citeable source of the dataset, credit is given to the data producer, which eliminates the concern about ownership by providing an official academic record of provenance.

As barriers for data sharing include concerns about loss of credit (Tenopir et al., 2020), the assimilation of sharing and reuse could be positively influenced by recognition (e.g. awards or other compensatory means, like co-authorship) of the expansion of scientific discovery through studies reusing available data (Piwowar & Vision, 2013; Curty et al., 2017; Tenopir et al., 2020). A way to recognise the producers of reusable data would be the connexion of an identifier for each researcher (e.g. ORCID) with an identifier for the dataset (e.g. DOI) which would then have to be cited in each new publication reusing the initial dataset (Piwowar & Vision, 2013; Pierce et al., 2019). In addition, publishers would have to ensure that these citations are available in a searchable system (e.g., Crossref) to establish a real link between data generator and publication (Pierce et al., 2019). Such a system, which recognises researchers regularly for generating data, could substantially influence the assessment of the value of scientific data by academic institutions, funders, and society (Pierce et al., 2019).

Establishing a reuse culture

Despite the reservations highlighted above many facets of data reuse provide incentives for the individual to practice it. Through open science initiatives (including but not limited to those listed in the Introduction), modern biologists are encouraged to make use of publicly available sequence repositories and mine data generated by others. Further, not comparing one’s dataset to publicly available analogues can be considered akin to ignoring replicated experiments (Denk, 2017). In fields where scientific progress has immediate and measurable positive impacts, such as medicine, the benefits of data reuse to society quickly outweigh the risks (Safran, 2017). Due to these advantages, there is an argument to be made that data reuse is an ethical obligation in the life sciences.

The statistic that the metagenomes of 20% of papers published between 2016 and 2019 are not publicly accessible (Eckert et al., 2020) demonstrates that there is still a long way to go until data sharing becomes routine. Therefore, open science incentives and database contribution guidelines should require the inclusion of metadata in all submissions to public datasets. Gamification via the implementation of fancy statistics about the data connected to a personal profile for each researcher might be another way to encourage researchers to share their data. A reusability score assigned by the community could increase the quality of the provided metadata. Such practices would not only encourage authors to collect data with reuse in mind (Goodman et al., 2014) but enable productive and valid re-analysis. Reinforcement could include spot-checking of provided metadata by funding agencies (Bhandary et al., 2018), as they have a monetary interest in the data quality of projects they support.

Only a combination of obligation and encouragement from the publishing and educational spheres are likely to ensure a future of successful and fair data ‘recycling’. The prevailing culture of positive publication bias (Mlinarić, Horvat & Šupak Smolčić, 2017) leads to ‘missing studies’ (the desk-drawer effect) and could also introduce a bias into analysis based on existing datasets (Wan & Pavlidis, 2007). For this reason, young researchers should be encouraged to share their data (if it is of sufficient quality) even if the outcome was ‘negative’. Additionally, bioinformatics should be integrated into the education of next generation life scientists to increase data re-use capacities (Soranno et al., 2015). As one learns about a new method of data collection so should one learn about the metadata that must accompany it in publication to render the data useful to others.

Examples of Successful Data Reuse

There are already numerous examples of successful studies from various areas of life science which involve intensive reuse of public datasets. Genomic data can, for example, be harnessed for pangenomic analyses (Montenegro et al., 2017) while transcriptomic and ChIP-seq data might be useful for the investigation or construction of regulatory networks (Chow et al., 2019). Phylogenetic analysis of groups from individual gene families (Du et al., 2016) to whole taxonomic groups (Bowles, Bechtold & Paps, 2020) benefits from reuse of genome, transcriptome and proteome data. Further, several tools and techniques have been developed for example mining antimicrobial peptides from public databases (Porto, Pires & Franco, 2017). The taxonomic classification of sequences identified in metagenomic studies is another application which heavily relies on available data as the quality of a study scales with the quality and size of the available data (Breitwieser, Lu & Salzberg, 2019). It can also be expected that machine learning will become even more ubiquitous in combination with other methods due to its ability to tackle large datasets and reveal novel patterns. Finally, in fields like modelling, the access to reusable data for the generation of models is even required (Curty et al., 2017).

In the life sciences, data reuse spans many data types and fields with substantial overlap in both categories. Table 1 shows selected reuse cases in the life sciences that cover many areas and concepts of data reuse sorted by the type of the analysed data. With every individual case of reuse, one must also consider the specific disadvantages associated with each approach. As highlighted above, there are risks to reuse. In addition to listing successful examples, Table 1 includes the limitations and risks associated with that particular method of reuse based on our assessment as data consumers and researchers. They illustrate the types of considerations that must be taken into account when reusing a specific type of dataset.

Table 1 Examples of dataset reuse for a novel purpose with the limitations/risks associated with each method.

Examples	Limitations/risks	
Genome	
Assembly of new genome sequences, for example organellar genome sequences, based on public datasets (Dierckxsens, Mardulyn & Smits, 2016)	Potential contaminations, for example of non-target organisms, are unknown. Only the submitter of the original reads can submit the assembly. There are several cases of contamination in published datasets as well as methods for the identification of such contaminations (Longo, O’Neill & O’Neill, 2011; Merchant, Wood & Salzberg, 2014; Strong et al., 2014; Delmont & Eren, 2016; Kryukov & Imanishi, 2016). One study found possible human DNA contamination in 72% of the analysed (n = 202) previously published metagenomes (Schmieder & Edwards, 2011).	
Motif identification, for example deep learning methods for identifying Poly(A) signals (Yu & Dai, 2020)	A large and suitable training set is required. A prediction accuracy of more than 90% can be achieved, but this highly depends on the context of the respective analysis.	
Pangenomic analysis, for example for bread wheat (Montenegro et al., 2017)	Assembly quality might differ between different studies due to factors like for example coverage. Samples with 10× coverage have an assembly efficiency of 81% using the IDBA-UD assembler (Peng et al., 2012; Montenegro et al., 2017), while high continuity long-read assemblies require at least 20–50× coverage (Lu, Giordano & Ning, 2016; Koren et al., 2017; Schmidt et al., 2017; Solares et al., 2018).	
GWAS to associate variants (QTLs,
SNPs) with traits, for example single-plant GWAS for identification of plant height candidate SNPs (Gyawali et al., 2019)	A large number of false positives requires large datasets, their sharing and compulsory replication (Marigorta et al., 2018). One possibility to check for sufficient sample size in for example genetic association studies is the random division of the study population by two and the requirement that any results have to be detected in both subsets (Hirschhorn et al., 2002).	
Transcriptome	
Co-expression analysis to find connected genes, for example identification of long non-coding RNAs associated with atherosclerosis progression (Wang et al., 2019); Co-expression networks, for example related to bamboo development using public RNA-Seq data (Ma et al., 2018) or related to cellulose synthesis using public microarray data (Persson et al., 2005);
Construction of regulatory networks using co-expression data, for example co-expression network analysis to reveal genes in growth-defence trade-offs under JA signalling (Zhang et al., 2020)	Batch effects might be possible if large sample groups come from the same source. Ideally, networks for different samples should be incorporated as there is high variation between co-expression networks with different samples (Ma et al., 2018).	
Gene expression analysis to find/identify best gene candidate for cloning (and select the right tissue), for example integration with GWAS to identify causal genes in maize (Schaefer et al., 2018)	Batch effects if large sample groups come from the same source. The success depends on the gene expression data context.	
Identification of qRT-PCR reference genes (Kwon et al., 2009; Cheng et al., 2011; Hruz et al., 2011)	Batch effects if large sample groups come from the same source. For example for formalin-fixed paraffin-embedded tissues, accurate normalisation requires two to four endogenous reference genes (Kwon et al., 2009). Further, for the normalisation of RT-qPCR data condition-specific reference genes should be used (Hruz et al., 2011).	
Gene prediction via analysis of RNASeq data (Pucker, Feng & Brockington, 2019) and for example GeMoMa is using this heavily (Keilwagen, Hartung & Grau, 2019)	Batch effects if large sample groups come from the same source. In regions without RNA-Seq data, ab initio prediction is required (Testa et al., 2015).	
Gene expression web sites, for example the eGFP browser (Winter et al., 2007)	Only genes in the annotation included. Only based on the available structural annotation thus alternative transcripts would be missed.	
Analysis of non-canonical splice sites based on genome sequences, annotations, and RNA-Seq datasets (Pucker & Brockington, 2018; Frey & Pucker, 2020)	Batch effects if large sample groups come from the same source and annotation errors will impact analysis results. One example is the high number of annotated CT-AC splice site combinations in fungal genome sequences which are probably caused by a systematic error in the assignment of RNA-Seq reads to DNA strands (Frey & Pucker, 2020).	
Extraction of new sequences for phylogenetic analysis (Schilbert et al., 2018)	Reliability of source is crucial; transcriptome assemblies are inherently incomplete as not all genes are expressed at the same time.	
Reuse of microarray data for meta-analyses including the investigation of non-coding RNAs (Raju, Tsinoremas & Capobianco, 2016)	Microarray technology is not comprehensive for example in comparison to RNA-Seq; the study is limited to known non-coding RNAs and is not suitable for the detection of new non-coding RNAs (Raju, Tsinoremas & Capobianco, 2016).	
Gene expression analysis based on microarray data (Wan & Pavlidis, 2007)	Submitters might fail to indicate technical replicates. Sequences of probes are sometimes unknown.	
Investigation of the underlying mechanisms of homeostatic eosinophil gene expression (Grace et al., 2018)	Variation in the methods for RNA-Seq library construction likely contributes to part of the detected differential expression (Grace et al., 2018).	
Proteome	
Identification of antimicrobial peptides (Porto, Pires & Franco, 2017)	Prediction, correct modelling and structural analysis are not completely accurate due to for example the presence of precursors. Validation is required.	
Phospho-proteomics, for example compartmentalisation of
phosphorylation motifs (Van Wijk et al., 2014)	Meta-analysis allows extrapolation only for highly specific conditions due to numerous different experimental conditions in the used studies. For example in seedling/rosette samples, plastid proteins might be (50%) overrepresented/mitochondrial and secretory proteins might be (10%) underrepresented in comparison to cell cultures/root/pollen/seed samples (Van Wijk et al., 2014).	
Metabolome	
Metabolic modelling (Brinkrolf et al., 2018)	Precise conditions of experiments are different between labs and measurement biases are possible.	
Combining network analysis and machine learning to predict
metabolic pathways (Toubiana et al., 2019)	Cannot be used to predict catalytic activity, but only to predict pathways. Stabilized correlation and reduced error rate can be achieved using a large sample size. Large sample sizes can be used for the exploitation of the natural variability of mapping populations or collections of different varieties or cultivars (Toubiana et al., 2019).	
Phenotype	
Deep learning methods for image-based phenotyping, for example leaf counting (Ubbens et al., 2018) or root and shoot feature identification (Pound et al., 2017)	Large datasets are required. For vision-based deep learning analyses, tens of thousands to tens of millions of images might be required (Ubbens et al., 2018).	
Ecology	
Modelling and prediction of the variability of biodiversity to explain ecological and evolutionary mechanisms (Jetz, Fine & Mace, 2012)	Choice and accuracy of predictor variables are crucial for the model: Challenges regarding the definition of exact region boundaries, climate reconstruction and comparability across clades remain; (Jetz, Fine & Mace, 2012).	
Ecosystem modelling, for example reuse of model code/reuse of eutrophication models for studying climate change (Mooij et al., 2010)	Partly overly simplified models: the validity of outcomes must be tested. Observations of species are sometimes placed at institutes of districts/regions.	
Database of lake water quality (Soranno et al., 2015)	Underlying datasets can be incomplete (e.g. missing lake coordinates)	

Assessment of Reuse Suitability for the Selection of Datasets

We have seen that in the selection of appropriate datasets for reuse, limitations and potential errors must be considered, in order to tap into the full potential of the practice, while avoiding invalid analyses. It has previously been demonstrated that a posteori analysis of dataset quality is possible with the quality control metrics for proteomics datasets to assess their suitability for a particular reuse purpose (Foster et al., 2011). Here, we provide a checklist (Table 2) to aid in the selection of datasets suitable for reuse, including suggestions, suitable controls and questions to consider prior to the re-analysis of public data. The questions are inquiries that a life scientist might consider when assessing a dataset of unknown quality and were determined by the authors.

Table 2 Checklist for the selection of appropriate datasets.

For each possible criterium, several questions to consider and suggestions for the reuse of public data are mentioned.

Criteria	Question(s) to consider	Suggestions/Suitable controls	
Integrity of the source	Is the source/submitter associated with data fabrication/plagiarism?	Check potential conflicts of interests/funding (useful resources: NSF conflict of interest guidelines (https://www.nsf.gov/pubs/policydocs/pappguide/nsf16001/aag_4.jsp), examples and strategies of dealing with conflicts of interest (Resnik, 2007)	
Biases	How was the data generated? Are there batch effects?	Comparison of random samples from the dataset with replacement (bootstrapping) to reveal any bias/errors; Principal component analyses	
Missing metainformation (sparsity)	Do you have all relevant information (e.g. information about the biological material)?	Possibility to contact the authors; infer metadata from datasets for example identify RNA-Seq tissue based on gene expression patterns of marker genes	
Integration of datasets from different sources	Is the data comparable?	Check relevant parameters:
For sequencing reads: same (NGS)
technology/platform and same sequencing chemistry (differences between versions of sequencing chemistry are possible)	
	Are the methods used for data
collection/generation comparable?	For assemblies: same type/version of bioinformatic tools and a full list of parameters	
Quality issues	Is the quality high enough to reach your goals (e.g. looking at gene expression differences between strains or making evolutionary trees)?	Check relevant parameters:
For sequencing reads: Phred scores, length, paired-end status	
	Are there any scores/hints available to check the quality of the dataset?	For assemblies:
continuity, contig/scaffold N50, auN (Li, 2020)	
Copyright/Legal issues	Are there any restrictions for reuse and publication of the data, especially due to the Nagoya protocol?	Check copyright information/licenses when selecting data prior to the actual reuse	

Conclusions

Data reuse is quickly becoming a ubiquitous part of research in the life sciences and scientists increasingly recognise the benefits of open reusable data (Tenopir et al., 2020). There are different steps to achieve and develop actual data-sharing behaviour which complies with ‘open data’ principles. As stated above (Fig. 1), technological progress together with changing research behaviour make ‘open data’ and its reuse (1) possible, (2) easy and (3) desirable. Considering the increasing quantity of available public data, in silico analyses are starting to supersede classic ‘wet lab’ experiments in some areas. However, it is still difficult to determine on a case-by-case basis whether the cost-benefit analysis favours data reuse with the associated risks or the increasingly cheaper/faster sequencing.

One of the factors promoting reuse behaviour would be the demonstration of its value (Curty et al., 2017). The provided list of successful examples in Table 1, which is only a narrow selection of studies conducted reusing publicly available datasets, illustrates the high potential value of data reuse. General limitations of these example studies include batch effects, quality issues and incomplete accuracy of predictions due to missing parameters (Fig. 4). Ultimately, the data itself, necessary to gain new scientific knowledge, is already available and only ‘awaits’ to be extensively investigated to answer open scientific questions.

Figure 4 Advantages and limitations of data reuse.

The reuse of publicly available scientific datasets leads to a reduction of costs and saves time, encourages reproducible research, enables the detection of novel information and has benefits for authors themselves (Fig. 4). Across the life sciences, there still remain some outstanding questions and challenges (Fig. 5). Considering all the advantages and taking into account the limitations we highly recommend and encourage data reuse when one is confident in that the reuse can be categorised as fair. We believe that the discussion of responsible data reuse must become more common in the life sciences so everyone can benefit from the largely untapped data resource.

Figure 5 Summary of outstanding questions and challenges.

Supplemental Information

Supplemental Information 1 Results of the literature review.

Click here for additional data file.

We are grateful to Katharina Schiller for helpful comments on the manuscript.

Additional Information and Declarations

Competing Interests

Author Contributions

Data Availability

The authors declare that they have no competing interests.

Katharina Sielemann analysed the data, prepared figures and/or tables, authored or reviewed drafts of the paper, and approved the final draft.

Alenka Hafner analysed the data, prepared figures and/or tables, authored or reviewed drafts of the paper, and approved the final draft.

Boas Pucker analysed the data, prepared figures and/or tables, authored or reviewed drafts of the paper, and approved the final draft.

The following information was supplied regarding data availability:

This article is a literature review and does not involve raw data or code.

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
