# Peer review of "The reuse of public datasets in the life sciences: potential risks and rewards"

_PeerJ, doi:10.7717/peerj.9954_

## Round 0.1 · original submission · Major Revisions

Dear Colleagues,
Both referees provided excellent reviews. Clear and useful and in full agreement of one another. Please consider their suggestions and concerns.

Reviewer 1 ·

Basic reporting

I believe this review is of broad interest. Although the scope is for life sciences, I believe there is wide applicability. Many of the concepts raised by the authors could be applied to other disciplines.
Reviews on open data have been published recently, but this manuscript stands apart for its discussion on some of the challenges and limitations they describe. I do not know of any review of this topic for life sciences so it definitely has a different audience. I believe this article to be timely and relevant.
The authors have a great command of the literature. They provide sufficient background although organization is a bit difficult. The introduction I found did not accurately set up the goals of the synthesis. I believe the first two paragraphs could be summarized more succinctly than currently written. The main point of these paragraphs is that big data is expanding due to technologies and initiatives. Revise to be more direct. Then I believe it would make sense to move lines 131-143 into the introduction set up for the reader the benefits of potential data reuse. That better ties in with your current paragraph three 56-64 that identifies there can be risks or negative impacts. Your final paragraph in the introduction then lays out the objectives.

Experimental design

The authors review is extensive and covers a broad number of articles. I believe the authors conducted a systematic review to have an unbiased approach in their writing, although the purpose of the systematic search was unclear. I would appreciate if the authors could provide some description of how their search to help generate the review.
The manuscript would benefit from begin structured around the end of their introduction. It is unclear in the main objectives of the synthesis is to address the three points on lines 67-69 or the “highlights” of challenges/limitations on lines 73-74. Explicitly stating the objectives of the introduction would assist, and then the rest of the review can be structured around that. Ensure the rest of the MS parallels these objectives.

Validity of the findings

I think the benefits section could be expanded on. In your figure 4 you highlight the benefits outweigh the costs, but your “challenges and limitations” section is considerably longer than your “benefits” section. I’m not imply these sections need to be equal, but I did feel the benefits section fell short on selling all the truly beneficial things associated with data re-use. For example, the prevent information loss section could be split to include a short discussion on novel analyses (e.g. meta-analysis, systematic reviews) that have incredible value in the life sciences. Perhaps even discussing the reuse of data from exploratory/preliminary experiments to reduce duplicate labour. This information is present in some forms throughout your MS, (e.g. lines 152-153) but I believe each could be expanded on.
I really enjoy the conclusions set up by the authors including figures 4-5.

Additional comments

Some specific comments
It would be good to end the abstract with big-picture statement, what does this paper do to advance this field
Line 56 – Don’t lead the paragraph with “However”. Needs a proper topic sentence.There are other instances throughout the MS where the authors start a new paragraph that is essentially a continuation of the previous. Please correct these.
Line 98 – I don’t understand this qualifier. Yes, your data exploration is only using life sciences, that was expected based on the title and abstract.
Line 106 – Some other examples to consider: code, statistics, and plots.
Line 117 – Define primary vs derived.
Line 218 - I am not sure what is meant by denormalization. It should be defined
Lines 268-270 - Combine these paragraphs
Line 88-89. This line is very vague for explaining how the information was obtained from the studies. Please expand. Similarly lines 90-91, what is the definition of a successful data reuse.
I believe table 1 could be moved to an appendix since it simply a series of examples.
Figure 1: I don’t understand the possible -> habitat gradient in figure 1.

·

Basic reporting

This review paper suggests that the volume and velocity of data in the life sciences creates opportunities and challenges for data reuse to enable novel analyses and improve reproducibility and accountability. In essence, the authors seem to be making a case for increased data reuse. As a conservation scientist I am not able to comment on the discipline-specific challenges of the genomics/bioinformatics data on which these authors focus, but as a proponent of open science whose work focuses on synthesis of diverse datasets generated by others, the general challenges around open science and open data discussed in this paper are very familiar to me. Unfortunately, I feel the organization of this paper makes it difficult to extract a coherent message or action that the reader, regardless of scientific discipline, can put into practice.

The stated intent of the study is to highlight the potential of data reuse (line 21, lines 24-25), with a secondary goal of motivating action on the part of the reader: “These benefits for individuals and the scientific community as a whole make a strong case for the obligation to reuse data whenever that is appropriate.” (lines 142-143) Considering the paper as a call to action, rather than simply a passive collection of loosely connected facts, it becomes important to keep a well-defined audience in mind so that the persuasive argument can be targeted toward the needs and incentives of the reader.

Here, a well-defined audience might be considered along two axes: targeted scope of discipline (general life sciences to specific communities within the life sciences), and the role of the target audience within the discipline. Re: scope of discipline: Is this intended for a general life sciences audience (as inclusion of phenology and ecology examples suggests), or a bioinformatics audience (as suggested by the bulk of the examples drawn from genomics sequencing literature, lines 126-128, as well as the subjects of Figures 2 and 3)?
• In attempting to address both these audiences, the conclusions drawn seem at once too vague to be useful to a bioinformatics practitioner and too domain-specific to be of use to a general audience.
• Other domains of life sciences have previously explored the space of “big data” in depth, (e.g. ecology, see Hampton et al. 2013: https://esajournals.onlinelibrary.wiley.com/doi/full/10.1890/120103) and should be acknowledged, especially if trying to appeal to a broader life science audience.
• The title suggests a general life sciences audience but the content focuses far more toward reuse of genetic sequence data. Assuming this is the target audience, I’d suggest a title change to reflect this focus.

Re: role of the target audience: Is this intended for data users/re-users/consumers, data producers, database managers, or policymakers? Any one of these audiences would be a reasonable target, but in attempting to address all at once, the discussion becomes unfocused.
• The recommendations and incentives for data reuse to each of these audiences would be quite different. For example, in choosing whether to reuse data, a researcher (data consumer) will consider only the benefits with respect to her particular research agenda, and would have no concern for benefits to a database manager elsewhere. See Lowndes et al. 2017 (https://www.nature.com/articles/s41559-017-0160, full disclosure I am a co-author) as an example of a piece narrowly focused on the needs and challenges of the data consumer, with concrete recommendations for actions the reader can implement.
• In the end, the decision to reuse or not reuse rests in the hands of the data consumer. The benefits to other stakeholders only accrues if the reuse of data actually occurs, therefore the considerations of other audiences should perhaps be focused on lowering the barriers and increasing the incentive/utility of reuse to the end user.
• The authors touch on several issues that would be of concern to the potential data consumer/reuser, including the need for high quality metadata (lines 261-274), clear and reproducible experimental design and methods (lines 209-210), and access to funding to incentivize/enable such reuse (line 313).

Experimental design

The outlined publication survey methodology seems like a reasonable approach to examine the scope of data reuse in public databases. However, the methods as written do not describe the process in enough detail to understand the study design or provide any means of replication.
• The authors state that in performing their keyword search, “the publication year and type of journal were unrestricted” (line 84). Since the noted keywords relate only to data, and not to discipline, the authors must have some other means of restricting the results to life sciences, if not even more narrowly, perhaps through additional keywords or a subset of type of journal.
• The authors state: “From these sources the common benefits, potential, limitations, challenges and risks of data reuse we extrapolated” (lines 88-89). Please provide reasonable description of the methods by which the returned journal articles were qualitatively or quantitatively analyzed to identify benefits, potential, limitations, challenges, and risks. It would be helpful to highlight/summarize the results of such an analysis in the text or in supporting materials.

In providing examples of “successful data reuse” drawn from different life sciences fields, it would be very helpful for the authors to clarify both their definition of “reuse” as well as their metric of “success” – which may influence and in turn be influenced by the intended audience.
• It may be that the issues of reuse that concern the authors are specific to the sequence-oriented life science community (for example, in conservation ecology, much of the research focuses on synthesis of existing datasets), which would significantly narrow the potential impact of this paper.
• Does “successful reuse” indicate publication of a study that incorporates data generated elsewhere (as seems to be the case from the examples given), or is there some way to quantify the degree to which reuse materially improves the utility of the study? Are there examples of “unsuccessful” data reuse that can highlight the challenges?

Validity of the findings

The authors lay out several potentially important challenges and benefits from data reuse, but the conclusions are not clearly supported by their evidence and reasoning as presented; what seems to be a central piece of their analysis, the publication survey, is not described or summarized in sufficient detail to support their defined list of risks, benefits, and potential of data reuse. Much of the paper (including Tables 1 and 2) seems to focus on the problems, warnings, and pitfalls of data reuse, rather than providing a convincing argument in favor. In conjunction with some of the organization and definition of terms and audience noted above, the authors will be better able to connect the goals of the analysis, as set out in the introduction, to their conclusion, with a clear and explicit chain of evidence-based reasoning.

Some specifics:

Table 1 seems to focus on the specific limitations of each reuse case example, which may be idiosyncratic - not be immediately applicable to other studies even within the same field.
• How were the “limitations/risks” in this table determined? E.g. were these identified by the authors of these studies that reused the data, or by followup studies analyzing or criticizing the reuse publication, or identified by the authors of this paper? Many of the limitations/risks fail to cite references.
• It would be particularly valuable to identify trends, patterns, and generalizable lessons from these examples that would be more widely applicable and useful to the targeted audience.

Table 2: the criteria column tracks well with the manuscript.
• As for table 1, were the “questions to consider” and “suggestions/suitable controls” determined from the literature? Few of the suggestions and none of the questions cite a reference.
• If these are the opinions of the authors, that should be made clear in the body of the manuscript or the caption of the table.

Figure 1: the text first referencing Figure 1 (line 31) does not clearly track the flow of the diagram – it seems like only boxes 1 through 3 are referenced here; it may be that the other steps are infused throughout the remainder of the paper, but this expectation could be explicitly signposted here to help the reader track the logical flow of the argument. Perhaps this figure could form a framework for a more clear outline of the piece. (Note also the caption seems incomplete).

Figure 3 seems extraneous – the main takeaway is a qualitative observation of exponential growth, which is described well within the text. In addition, when the authors argue that “the increasing size of scientific data is a challenge for the databases regarding e.g. storage space and data management” (lines 172-173), I think it would be helpful to cite evidence that this exponential growth of data is outpacing the (likely) exponential growth of affordable access to storage and processing power.

---

## Round 0.2 · accepted · Accept

Dear Colleagues,

Thank you for taking the time to address all issues.

Best,
Christopher.

Reviewer 1 ·

Basic reporting

No comment

Experimental design

No comment

Validity of the findings

No comment

Additional comments

This is my second time reviewing the paper by Sielemann et al. I commend the authors on all the revisions they undertook to improve the clarity and reading of the manuscript. I believe these revisions have made for a considerably stronger manuscript and enjoyed reading it. Thank you for addressing all my comments.

I have only a few small outstanding issues:
-Try and reduce your usage of "it". Sometimes it comes across what "it" you are referring to. Just restate the subject of the sentence again
- There are a couple cases where you are using double parentheses. In these cases you can simply separate the citation from the example statement using a semi colon. (e.g. line 306)
- Table 1 is very dense. Can you separate it a bit more. Perhaps adding the citations to a third column
- I don't really understand the usage of these colours in Figure 1. Could this instead be grey and white? with the boxes outlined rather than shaded in. I believe it will improve the aesthetic.

·

Basic reporting

This review paper outlines the needs, benefits, and challenges of data reuse in the biological sciences, with examples drawn from a range of biological applications. The considerable effort put in by the authors to clarify and organize the paper have strengthened the manuscript considerably.
The language is generally clear though there are a few typos and odd turns of phrase to be addressed (listed below). The intro and background have been clarified to provide context to the goals of the literature review. Literature is thoroughly and appropriately cited throughout. The structure conforms to PeerJ standards, and the authors use helpful subheadings throughout to communicate the structure of the piece. The review fits within the scope of the journal, and while focused on particular disciplines the general takeaways are generalizable to other disciplines. The field may have been reviewed recently, but in a quickly evolving topic like big data and open science, it seems like there is good reason for this review. Finally, the introduction has been reworked to better communicate the authors’ goals, particularly with the “purpose of the review” subsection.

Some specific suggestions:
Lines 75-77: “Additionally, a survey of 100 datasets associated with ecological and evolutionary research showed that 56% databases were incomplete and 64% were archived in a way that partially or entirely prevented reuse” – this is great, but you may also want to check out this reference that I think is very relevant to the point you are making in this paragraph:
• Couture, J. L., Blake, R. E., McDonald, G., & Ward, C. L. (2018). A funder-imposed data publication requirement seldom inspired data sharing. PLOS ONE, 13(7), e0199789. https://doi.org/10.1371/journal.pone.0199789

I noticed a few typos and awkward phrases that the authors may want to address:

Lines 41-43: “Obligations and benefits lead to established sharing behaviour and more datasets become available which can then be reused in turn”: This reads as a run-on sentence. Perhaps something like “Obligations and benefits lead to established sharing behavior, which leads to more datasets becoming available, which can then be reused in turn”

Lines 44-46: “Data sharing is already common in several disciplines, including 
genomics, neuroscience, geoscience and astronomy and an increasing number of studies reuse shared data”: This reads as a run-on sentence. Perhaps a comma after “astronomy” might help.

Line 112: “life scientist”: Should be “life scientists”?

Line 120: “risks and potential solutions range cross different fields”: Should be “range across different fields”?

Line 160: “like e.g. ‘coexpression’, ‘pangenomcis’, ‘network analysis’ or ‘metagenomcis’.”: “pangenomics” and “metagenomics”? also drop the “like,” as it is redundant with the “e.g.”

Line 282: “can be partially addresses”: “addressed”?

Lines 299-301: “As discussed above, open access to datasets and studies would accelerate science while being 
cost-efficient (Spertus, 2012), however, it is important that the limitations of particular datasets are identified, and the associated risks assessed.”: Semicolon or period before the “however,” rather than a comma. I think the comma after “identified” can also be dropped.

Lines 390-391: “Re-analysis of publicly available data is one way to tackle the issues of unknown quality and denormalization of data, and integration of databases that arise with reuse”: Maybe “…unknown data quality, denormalization of data, and integration…”?

Line 420: “The trade-off between public access and unknown quality, can be partially resolved”: No need for the comma.

Line 458: “data-focused journals, like e.g. GigaScience”: Drop “like” – it is redundant with “e.g.”

Lines 496-497: “The above-mentioned ‘Parasite awards’ use this tongue-in-cheek name after it was introduced to dismay such practices”: Not sure what this means – “to dismay such practices” – perhaps a different verb was intended?

Line 510: “Both, data providers and creators of databases, deserve recognition for their work”: Neither comma is needed.

Lines 553-555: “Additional enforcement could be spot-checking of provided metadata by funding agencies (Bhandary et al., 2018) which are providing the money for the data generation and have an interest in the release of high quality data sets”: I can see what the authors are intending but the sentence is very awkward.

Line 557: “Only combination of obligation and encouragement”: “Only a combination…”?

Line 591: “They aim illustrate the types of considerations”: “They illustrate the types…”?

Line 598: “It has previously been demonstrated that posteori analysis”: “… a posteriori analysis”?

Line 619-620: “illustrates the high potential of data reuse, thus demonstrating its value”: Seems somewhat redundant - I think this could be shortened to “illustrates the high potential value of data reuse.”

Experimental design

I appreciate the additional information the authors provided about their methods for the literature survey. While the survey methodology may not be comprehensive or unbiased, the authors are transparent about their selection criteria so it should not be a concern.

For the examples listed in Table 1, I still note that many of the ideas listed under limitations and risks are not cited, which is not a deal breaker but I think weakens the authors’ point here. I understand that many of these are drawn from the researchers’ experience in the field, but some assertions are quite vague or speculative. I would urge the authors to find a citable source for each of these noted risks to strengthen their assertions; even if the ideas are drawn from the researchers’ experience, certainly they have been documented elsewhere in an academic manner. For example, one risk noted multiple times is, “Batch effects might be possible if large sample groups come from the same source” – there seem to be many publications and textbooks with information on batch effects, e.g. https://www.researchgate.net/profile/Qian_Liu113/publication/304497610_Evaluation_of_Methods_in_Removing_Batch_Effects_on_RNA-seq_Data/links/577175d208ae0b3a3b7d6f96.pdf.

I have the same concern about the suggestions/suitable controls in Table 2.

Validity of the findings

With the extensive reworking by the authors, their argument seems much better developed and supported. The conclusion is generally supported by the points made in the discussion, and links back to the goals set out in the introduction.

That said, it seems unusual to introduce figures for the first time in the conclusion. I would introduce Fig. 4 in the discussion section, and use that to tie together all the pieces of the discussion before moving on to the conclusion. While the conclusion is an appropriate place for “outstanding questions and challenges,” as you outline in Fig. 5, it seems like those questions should really focus on potential research opportunities going forward. I would suggest dropping Fig. 5 (or moving it to the discussion) and instead craft one good concluding question/challenge that could tie together some of these ideas. One major challenge that is telegraphed within your text would be identifying opportunities to align incentives among data users, data creators, and other stakeholders (e.g., database managers, funding institutions, journals, academic institutions) to better support effective data sharing.

Additional comments

Overall, I appreciate the work the authors have put in to address previous concerns, and find the paper to be greatly improved in its organization and message.